# SortedRL: Accelerating RL Training for LLMs through Online Length-Aware Scheduling

## Abstract

Scaling reinforcement learning (RL) has shown strong promise for enhancing the reasoning abilities of large language models (LLMs), particularly in tasks requiring long chain-of-thought generation. However, RL training efficiency is often bottle-necked by the rollout phase, which can account for up to 70% of total training time when generating long trajectories (e.g., 16k tokens), due to slow autoregressive generation and synchronization overhead between rollout and policy updates. We propose *SortedRL*, an online length-aware scheduling strategy designed to address this bottleneck by improving rollout efficiency and maintaining training stability. SortedRL reorders rollout samples based on output lengths, prioritizing short samples forming groups for early updates. This enables large rollout batches, flexible update batches, and near on-policy micro-curriculum construction simultaneously. To further accelerate the pipeline, SortedRL incorporates a mechanism to control the degree of off-policy training through a cache-based mechanism, and is supported by a dedicated RL infrastructure that manages rollout and update via a stateful controller and rollout buffer. Experiments using LLaMA-3.1-8B and Qwen-2.5-32B on diverse tasks, including logical puzzles, and math challenges like AIME 24, Math 500, and Minerval, show that SortedRL reduces RL training bubble ratios by over 50%, while attaining 3.9% to 18.4% superior performance over baseline given same amount of data.

## 1 Introduction

Large language models (LLMs) have achieved remarkable performance across a wide range of tasks (Achiam et al., 2023; Yang et al., 2025; Gemini; Liu et al., 2024). Recently, reinforcement learning (RL) has emerged as a key methodology for further enhancing the abilities of pretrained LLMs (Jaech et al., 2024; Guo et al., 2025; Seed et al., 2025). Notably, several works have focused on improving the reasoning abilities of LLMs on complex tasks such as mathematical problem solving (Hendrycks et al., 2021; He et al., 2024) and competition-level coding (Jain et al., 2025). These methods typically instruct the model to generate intermediate chain-of-thought steps alongside the final answer, and apply RL using outcome-based rewards (e.g., correctness of the final answer) as supervision. Concretely, the RL training procedure alternates between a generation step and an updating step. In the generation step, the model produces rollouts, comprising both the reasoning process and the answer, given an input question. In the updating step, the model parameters are updated based on the rewards assigned to the generated rollouts. This paradigm has shown substantial improvements in the reasoning performance of LLMs (Guo et al., 2025; Team et al., 2025).

A key observation in RL for LLMs is that improvements in model performance during training are often accompanied by an increase in the length of generated responses. This phenomenon is commonly attributed to the model's emergent ability to learn more complex reasoning, and suggests that encouraging the generation of longer responses, which often contains richer intermediate steps, plays a crucial role in enhancing overall performance.

However, because LLM generation is an auto-regressive process, producing long rollouts can be extremely time-consuming, making the generation phase the primary bottleneck in the overall RL training time. Moreover, because widely-used RL algorithms are all on-policy ones, whose training iterates between generation and updating, but the updating step can only begin after all responses in a batch have finished generating. When response lengths vary widely across fed samples, this

leads to inefficient hardware utilization (commonly referred to as "bubbles") as resources are left underutilized waiting for the longest generation to complete. Notably, we observe that the distribution of response lengths follows a long-tailed pattern (see Fig.1c), which exacerbates these bubbles and makes inefficient utilization especially pronounced during RL-based LLM tuning.

A straightforward way to mitigate those inefficiencies is to use large batches for rollout and employ optimizations such as continuous batching (Yu et al., 2022) or chunked prefilling (Agrawal et al., 2024) to reduce idling time and minimize the bubble. However, if the rollout batch is made large while keeping the update batch size fixed, the model would be updated multiple times using the same batch of generated data. This forces the model to train on increasingly off-policy data after each update, which can undermine training stability. On the other hand, forcing the update batch to match the rollout batch size is not a flexible approach and may not yield optimal results across different tasks. Thus, these naive solutions have inherent limitations that restrict their applicability in practical RL-based LLM training.

To address these challenges, we propose *SortedRL*, a method designed to reconcile sample and computation efficiency in RL-based LLM tuning. The first key component of SortedRL is online length-aware scheduling: it sorts incoming data by response length and prioritizes updates with samples that have shorter responses. This creates a micro-curriculum online without extra overhead. The updated model is then immediately used to generate rollouts for the remaining longer samples in the current batch, enabling large rollout batch size, flexible update batch size, and on-policy updates at the same time. Second, we introduce a mechanism to control the degree of off-policy training within the algorithm. By caching unfinished samples, SortedRL accelerates the pipeline further by enabling immediate processing of newly completed generations. Third, we develop a dedicated infrastructure to support SortedRL, abstracting the data generation and usage flow for RL training. This infrastructure features a length-aware controller to manage the rollout process and a stateful rollout buffer to dynamically coordinate the timing of model updates, thereby maximizing throughput and maintaining training consistency.

Extensive experiments demonstrate the effectiveness of SortedRL in both mitigating the bubble and mitigating negative impacts from off-policy training compared with the baseline. With Sorte-dRL, LLaMA-3.1-8B-Instruct attained the same high score using 40.74% fewer samples on logical reasoning tasks with vanilla Reinforce++ training. Using the same amount of training data, our approach exhibited 3.9% to 18.4% better performance on competition-level mathematical problems including OlympiadBench (He et al., 2024), AIME 2024, and AMC 2023. Meanwhile, via end-to-end performance test, we also observed a sharp drop in bubble ratio, from 74% to less than 6%.

## 2 MOTIVATION AND PRELIMINARIES

### 2.1 BACKGROUND

**RL for LLMs**   RL training for LLMs is a compute-intensive, multi-stage process characterized by heterogeneous components, unlike the uniformity in supervised finetuning. A typical RL pipeline consists of three key stages: (1) Rollout, where the actor model generates responses from input prompts; (2) Inference, where inference is performed using critic, reward, and reference models to compute values, rewards, and log-probabilities; and (3) Model update, where gradients are computed and applied. Scaling this pipeline has been shown to enhance LLM reasoning capabilities, especially in generating longer and more coherent Chain-of-Thoughts (CoTs).

**Proximal Policy Optimization (PPO) and Reinforce++**   PPO (Schulman et al., 2017) and Rein-force++ (Hu, 2025) are both REINFORCE-based policy optimization methods. Specifically, they update the policy by maximizing the following objective:

$$\mathcal{J}(\theta) = \mathbb{E}_{\substack{(q,a)\sim\mathcal{D}, \\ o\sim\pi_{\theta_{\text{old}}}(\cdot|q)}} \left[ \min\left( \frac{\pi_\theta(o_t \mid q, o_{<t})}{\pi_{\theta_{\text{old}}}(o_t \mid q, o_{<t})} \hat{A}_t, \ \text{clip}\left( \frac{\pi_\theta(o_t \mid q, o_{<t})}{\pi_{\theta_{\text{old}}}(o_t \mid q, o_{<t})}, 1 - \varepsilon, 1 + \varepsilon \right) \hat{A}_t \right) \right],$$

(1)

where, $(q, a)$ denotes a question–answer pair sampled from the data distribution $\mathcal{D}$, $\varepsilon$ is the clipping range for the importance sampling ratio, and $\hat{A}_t$ represents an estimator of the advantage at time step $t$. The computation and normalization of $\hat{A}_t$ can vary across algorithms (Schulman et al., 2017;

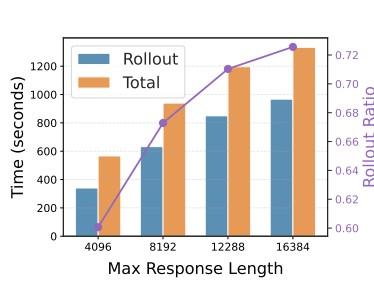 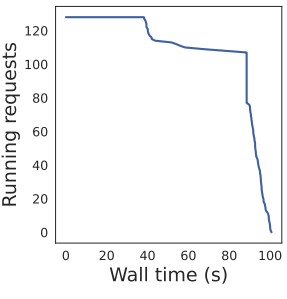 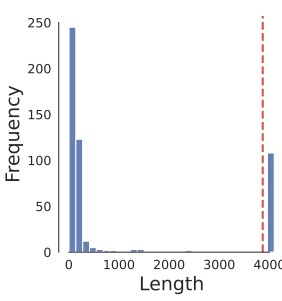

(a) Rollout is the bottleneck.     (b) Rollout bubbles due to sync.     (c) Variable output lengths in batch.

Figure 1: (a) Latency breakdown of RL training for LLMs. (b) GPU wall time per rollout batch with 128 batch size. (c) Length distribution of sampled trajectories during rollout. Visualizations are based on DeepSeek-R1-Distill-Llama-8B (Guo et al., 2025) with a 4K maximum generation length.

Shao et al., 2024; Liu et al., 2025; Hu, 2025). For example, the advantage functions used in PPO and Reinforce++ are shown in Eq.(2) and Eq.(3), respectively.

$$\hat{A}_{\text{PPO},t} = \sum_{l=0}^{T-t-1} (\gamma\lambda)^l \, \delta_{t+l}, \quad \delta_t = r_t + \gamma \, V_\psi(s_{t+1}) - V_\psi(s_t) \tag{2}$$

$$\hat{A}_{\text{Reinforce++},t} = \frac{R_i - \mu_{\text{batch}}}{\sigma_{\text{batch}}} \tag{3}$$

## 2.2 ROLLOUT DOMINANCE IN RL TRAINING COST

RL for LLMs encompasses three primary stages: actor model generation (rollout), evaluation via critic/reward/reference model inference, and subsequent actor/critic model training. Scaling RL training demonstrably improves LLM reasoning, notably by facilitating longer Chain-of-Thought (CoT) generation. However, this introduces a significant bottleneck: the actor model rollout increasingly dominates the training duration, as shown in Fig. 1a. For example, with a maximum generation length of 16K tokens, rollouts can consume 70% of the computational resources. For a state-of-the-art (SoTA) 32B parameter reasoning LLM, a single rollout step typically requires 15-20 minutes (Yu et al., 2025). As in most LLM-serving systems, autoregressive rollout throughput is primarily constrained by limited HBM bandwidth, due to frequent loading of model weights and KV caches. Moreover, the temporal dominance of rollouts is poised to intensify with more complex scenarios, such as agent-based interactions or multi-turn reasoning (Wang et al., 2025), which demand even longer trajectories.

## 2.3 BUBBLES IN ROLLOUT

A significant challenge stems from the considerable variance in generation lengths across different rollout trajectories. As illustrated in Fig. 1c, within a sampling batch size of 512, while 80% of samples are generated within 3K tokens, the remaining 5% can extend up to the token limit. This heterogeneity in generation length distribution is pervasive across all stages of RL training.

Critically, unlike standard LLM inference scenarios, the rollout model generation process is synchronous with actor model gradient updates. This synchronization precludes the application of continuous batching techniques (Yu et al., 2022), which are commonly employed in LLMs inference to optimize GPU resource utilization. Although previous works (Zhong et al., 2025; Mei et al., 2025) have sought to improve GPU utilization by stage fusion and overlapping rollout generation with actor model training, the efficacy of this approach diminishes with longer trajectories, leading to substantial GPU idle periods (i.e., large bubble ratios) during the rollout phase.

## 3 SORTEDRL

Following the analysis in §2, we propose SortedRL to reduce the bubble ratio in the rollout stage and accelerate RL training for LLMs. SortedRL comprises three key components: 1) Online Length-Aware Scheduling, a length-aware batching mechanism is employed to minimize rollout bubbles by aligning samples with similar generation lengths within a batch. 2) Controllable Off-Policy Sampling, supports both on-policy and partial off-policy modes, enabling flexible trade-offs between stability and sample efficiency during training. 3) Co-Designed RL Infrastructure, includes a length-aware controller and a stateful rollout buffer, designed to coordinate length scheduling, buffer management, and model interaction efficiently.

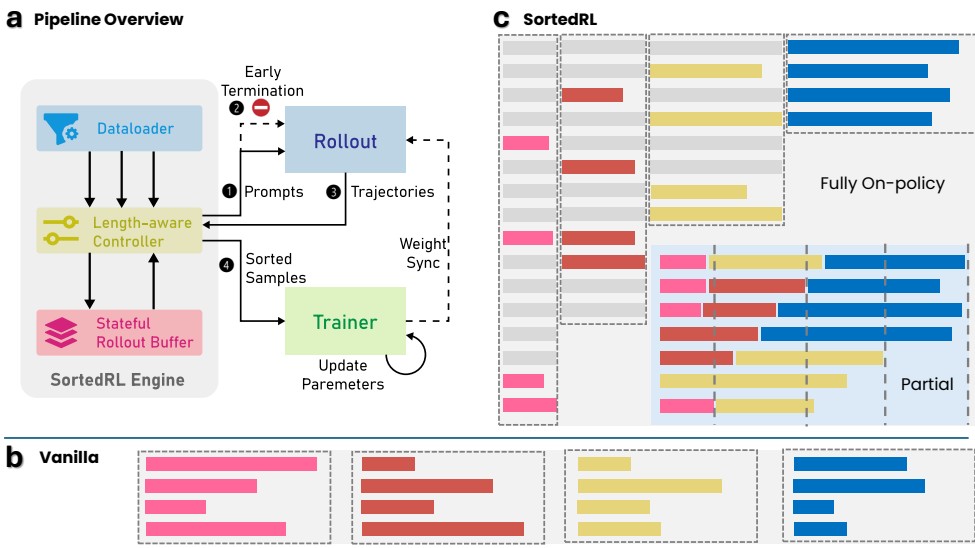

Figure 2: The SortedRL framework. **a**, The architecture of the SortedRL engine, consisting of two core modules: a length-aware controller and a stateful rollout buffer. The RL training pipeline includes five key steps: 1) concatenate buffer and feed prompts, 2) early termination, 3) collect and update rollout trajectories, and 4) sort and feed training batches; **b** and **c**, imaginary timeline and SortedRL strategy. Samples in same batch are denoted in same color. Dotted lines and boxes indicates the harvest timing. For fully on-policy mode, the gray bars are partially discarded incomplete samples or non-scheduled prompts, while there is no discarded trajectories in partial mode.

### 3.1 ONLINE LENGTH-AWARE SCHEDULING

To alleviate the prolonged rollout duration and high bubble ratio in RL training, we propose an online length-aware scheduling method. This approach dynamically batches samples with similar output lengths to minimize idle computation and reduce bubble ratios during rollout. To ensure token efficiency, we introduce the following key designs:

**Length-aware Rollout Schedule.** In the RL rollout stage, bubbles primarily arise from mismatches in generation lengths across samples. Long-tailed generation drastically slows down generation due to unsaturated device utilization under small batch size. Therefore, effectively reducing the bubble ratio hinges on accurately sensing generation length. Previous methods (Fu et al., 2024) have attempted to predict generation length using offline neural networks trained on specific datasets. However, these approaches often lack generalization to unseen scenarios and are ill-suited for reinforcement learning settings, which involve multi-sampling and continual updates.

To address this, we propose an generation-length aware rollout schedule that sense the fine-grained dynamics of generated sequences. Our controller adopts an oversubscription strategy, feeding the rollout engine with more prompts than its maximum queue capacity in most iterations. This ensures that the rollout engine consistently operates at its optimal batch size, as captured by hardware runtime

graphs (e.g., CUDA and HIP graphs), which is essential for maximizing the efficiency of JIT-compiled kernels.

In conjunction with oversubscription, the controller applies early termination based on batching-related thresholds. Once the condition is met, both completed and partially generated outputs are harvested. This mechanism effectively reduces idle time and minimizes computation bubbles during the rollout phase.

**Grouped Rollout and Micro-curriculum.** While oversubscription and early termination improve hardware efficiency, they also introduce a side effect: longer generations are more likely to be interrupted during training, resulting in a bias toward shorter responses in the collected data. Although partial generations can be scavenged and resumed in the next rollout iteration, the resumed segments are inevitably off-policy, and longer sequences are inherently suffering higher impacts from it due to more interruptions.

To mitigate this, we organize prompts into groups of batches and enforce a cache-aware loading policy: no new prompts are loaded from the dataloader until all cached prompts have been consumed. This strategy ensures that all prompts are fully processed within a bounded timespan, avoiding prompt starvation and maintaining balanced training dynamics.

As a result, SortedRL naturally batches responses of similar lengths together. Since shorter responses are typically completed earlier, outputs within a time slice tend to be temporally clustered, leading to length-sorted batching as the rollout progresses. Considering the length-reward correlation in reasoning models, batches in SortedRL groups naturally constructs micro-curriculum that has incremental difficulty. This sorting behavior can also substantially affects batch normalization, together contributing to improved token efficiency, which will be demonstrated and discussed in §4.

**Selective Batching for Training.** Unlike the canonical RL pipeline, our controller can selectively batch ready trajectories and feed them to the trainer in a dedicated order and combination. This is particularly important for algorithms such as Reinforce++, where batch-wise normalization can substantially impact training outcomes.

## 3.2 CONTROLLED OFF-POLICINESS

SortedRL supports two switchable modes of operation: **fully on-policy** and **partial** mode. As illustrated in Fig. 2, on-policy mode only feeds trajectories generated from the latest available policy, and terminates any unfinished requests in the queue. The prompt of terminated requests are scavenged back to buffer for rollout in next batch, to prevent starvation of certain prompts. Partial mode also scavenges generated tokens of terminated requests. The corresponding log probabilities used to generate the tokens are also cached in the buffer. In the incoming batch, the interrupted trajectories (prompts and generated tokens) are fed into the rollout engine. We concatenate log probabilities used to generate new sequence with the previously cached ones, to ensure every token can use the exact log probability value that was used to generate each token during importance sampling (Eq.1).

## 3.3 CO-DESIGNED RL INFRASTRUCTURE

For SortedRL, we co-design a compute-efficient rollout controller and a stateful rollout buffer to maximize MFU, support agile algorithmic experimentation, and maintain intermediate stateful results across rollout iterations. The underlying infrastructure provides the following key capabilities:

**Rollout State Manager.** The controller maintains a set of states, including: (1) unconsumed prompts from the dataloader, (2) scavenged response segments obtained upon rollout termination, and (3) the corresponding log probabilities for these segments. Scavenged segments can be concatenated with their original prompts to resume generation in subsequent iterations. Additionally, the stored log probability segments can be reused for importance sampling and serve as $\pi_{old}$ in Eq. 1 during policy updates.

**Stateful Rollout Buffer.** To support different modes in controllable off-policy training, we implement a stateful rollout buffer that stores intermediate results of partially generated trajectories.

Specifically, each entry in the buffer includes: the prompt context, the current partial trajectory, corresponding log probabilities, a completion flag indicating whether the trajectory is finished, and a lifecycle indicator used to determine when the entry should be cleared. This design ensures that partially generated trajectories can be efficiently resumed or discarded based on training mode and resource constraints.

# 4 EXPERIMENT

## 4.1 EXPERIMENT SETUP

**Dataset**    We evaluate our approach on two sets of data with distinct nature, and the ground truth data are suitable for rule-based evaluation:

First, a logical puzzle dataset, LogicRL (Xie et al., 2025). This dataset is composed of 5000 synthetic The Knights and Knaves game puzzles (Xie et al., 2024). An example can be checked in Fig.7, the game instructs players to deduce the roles of the characters mentioned in the statement. The training dataset is a mixture of 3 to 7 characters (i.e., different difficulties), with each difficulty accounting for 1000 samples. The samples are all shuffled during training. We spare 10% of data for evaluation.

Second is a mixed mathematical dataset, DAPO-Math-17k. This dataset contains a variety type of mathematical problems from the AoPS website. For easy and precise verification, the problems are transformed to expect an integer solution.

To evaluate the model's mathematical capability, we select 6 benchmarks following standard practice   (Yu et al., 2025; Shi et al., 2025; Zeng et al., 2025): GSM8k (Cobbe et al., 2021), MATH500 (Hendrycks et al., 2021), Minerva Math (Lewkowycz et al., 2022), OlympiadBench (He et al., 2024), AIME 2024, and AMC 2023. For problems from competition, considering its relatively small amount, we collect 32 responses for each question and record accuracy (mean@32).

**Models**    We select two scales of base model for two tasks. The model selection is determined by robust and reproducible baseline performance on specific downstream task. For example, we have observed that small model (Qwen-2.5-7B) is very limited in test-time scaling on math dataset, after an abrupt increment at the initial stage of RL training, which is possibly the process of learning the suitable format for rule-based verification, the model performance stopped improving (Fig.9b). Instead, Qwen-2.5-32B is free from such concern and exhibits robust and progressive improving pattern on training and evaluation metrics. As a result, for lightweight exploration on LogicRL, we employed LLaMA-3.1-8B-Instruct as base model, and Qwen-2.5-32B for mathematical problems.

**Compared Baseline**    For logical Reasoning, following the original practice (Xie et al., 2025), we train LLaMA-3.1-8B-Instruct on LogicRL dataset with Reinforce++. We set the rollout prompt batch size to 128 and collect 8 responses from each prompt, then update with a trajectory batch size of 1024.For mathematical tasks, we finetuned Qwen-2.5-32B on the DAPO-Math-17k dataset with PPO. The rollout prompt batch size, number of responses per prompt, update batch size are 512, 1, and 128, respectively. We adopt training tricks from DAPO (Yu et al., 2025) in both tasks including clip-higher, removing KL divergence. Meanwhile, we removed entropy loss for better stability.

**Implementation and Settings**    Our rollout scheme is implemented as a integrated component of VeRL, an open-source RL training framework (Sheng et al., 2024). Experiments in this work are conducted with SGLang (Zheng et al., 2024) as rollout engine.  Some patches are applied to facilitate better control of the rollout states and increase stability, including but not limited to: Incomplete request state retrieval, distributed communication timeout prevention, and platform-specific optimizations.  Our training was conducted on a mixture of NVIDIA H100 and AMD MI300X GPU cluster, equipped with 96-core Intel Xeon Platinum 8480C CPUs.

## 4.2 RESULTS ON LOGIC PROBLEMS

We first examine effects of our strategy on LogicRL dataset, on which we can obtain a stable learning pattern using vanilla methods. For better instruction following in logical games, we choose LLaMA-3.1-8B as starting point. Both baseline and two modes of SortedRL operates at a rollout batch size

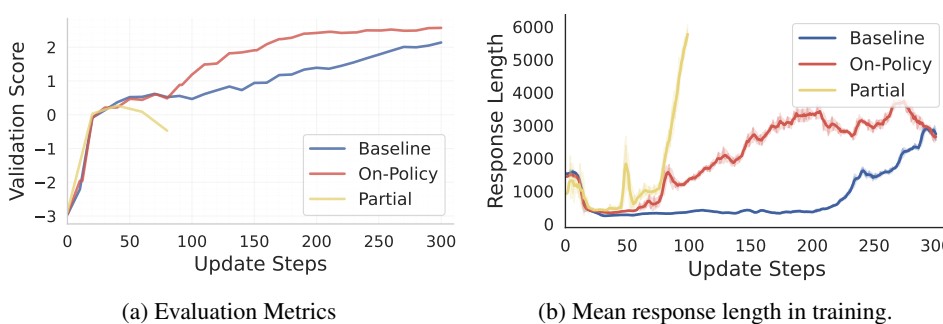

(a) Evaluation Metrics            (b) Mean response length in training.

Figure 3: LogicRL overall results. **On-Policy** and **Partial** are variants of SortedRL

Table 1: Evaluation results at $600^{th}$ checkpoint. **On-Policy** and **Partial** are variants of SortedRL

|  | GSM8K | MATH 500 | Minerva | Olympiad | AIME (mean@32) | AMC (mean@32) |
|---|---|---|---|---|---|---|
| **On-Policy** | 91.96 | **79.20** | **30.88** | **47.77** | **23.33** | **69.84** |
| **Partial** | 93.56 | 79.0 | 30.15 | 43.77 | 20.83 | 63.05 |
| **Baseline** | **95.15** | 76.2 | 29.04 | 44.51 | 19.69 | 63.13 |

of 128 prompts (8 responses per prompt) and update batch size of 1024 trajectories per step. For SortedRL, we chose fully on-policy mode to mitigate the off-policy update setting in hyperparameters.

The result are shown in Fig.3a. For both trials, the validation score underwent an sudden increase at initial 25 steps, where the model learnt to robustly capture the correct output format and avoided points deduction for format. At the same time, the mean response length also dropped to a minima. Subsequently, on-policy SortedRL rapidly improved in evaluation at around 80 steps, while baseline approach improves at a slower pace. Until achieving comparable evaluation score (for instance, 2 points), baseline method lagged for around 3 epochs. Interestingly, the leading behavior is also exhibited in the response length pattern. In SortedRL, the model started to explore lengthy responses 150 steps earlier than baseline. In contrast, the off-policy partial training shortly encountered a explosion in reasoning length, which finally fell into an unrecoverable degradation in performance.

### 4.3 RESULTS ON MATH PROBLEMS

To explore the effects of SortedRL in more challenging tasks and identify the impact of off-policiness, we finetuned Qwen2.5-32B on DAPO-Math-17k dataset. As a baseline, we set rollout batch size to 512 and update batch size to 128, resulting in 4 off-policy updates in each iteration. For both modes of SortedRL, we set the rollout batch size to 128 and group size of 512, with a update batch size of 128. This setup indicates 4 on-policy updates and 4 semi-off-policy updates in each iteration, respectively for on-policy and partial mode.

After 600 update steps, baseline, on-policy and partial SortedRL achieved 23.33%, 20.83%, and 19.69% accuracy (mean@32) in AIME24 (Fig. 4). Furthermore, we took the checkpoint at the $600^{th}$ update step to evaluate each approaches' performances on an ensemble of benchmarks. Results are shown in Tab.1.

The evaluation metrics and curve clearly demonstrates that following the decreasing order of off-policiness (on-policy - partial SortedRL- baseline), the token-efficiency increases accordingly. This aligns with our expectation that off-policiness negatively impacts token-efficiency, and partial SortedRL perfectly sits between fully off-policy baseline and on-policy SortedRL. And performances on MATH500, Minerva, Olympiad, and AIME24 follows the off-policiness order. However, it is intriguing that GSM8K seems being inversely impacted from the off-policy training.

Given the effect of off-policiness, we showed that SortedRL provides two relatively on-policy methods to conduct RL LLM training, yet ensuring the system operates at the maximum batch.

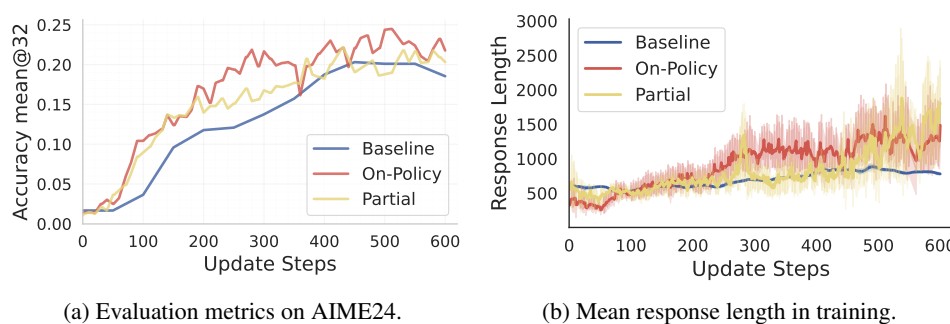

(a) Evaluation metrics on AIME24.

(b) Mean response length in training.

Figure 4: Mathematical task overall results. **On-Policy** and **Partial** are variants of SortedRL

## 4.4 ANALYSIS

### 4.4.1 THROUGHPUT OF DIFFERENT METHODS

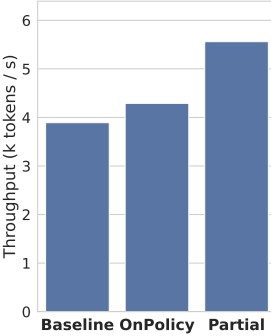

Figure 5: Rollout throughputs under different strategies.

We examined the end-to-end rollout efficiency of our underlying infrastructure by testing throughput of the different methods under the workload of 512 samples in 4 separate batches with a maximum generation length of 8k. To avoid non-deterministic behavior in generation, we set the sampling parameters for each sample to let generation lengths be exactly the same as baseline.

The results (Fig.5) are 3987, 4289, and 5559 output tokens per second respectively for baseline, fully on-policy mode and partial mode. Equivalent to a speedup of 7.57% and 39.48% for fully on-policy mode and partial model, respectively. We define a bubble ratio in eq.4, where $Q$, $T$, $r_k$, $t_k$ denote running queue size, total elapsed time, running requests, and duration, respectively. Compared to baseline bubble ratio of 74%, on-policy and partial mode reduced the number to 5.81% and 3.37%.

$$\text{Bubble Ratio} = \frac{\sum (Q - r_k)\,\Delta t_k}{T\,Q}, \qquad (4)$$

### 4.4.2 ABLATION STUDY ON KEY DESIGNS

We conducted a set of ablation study (Fig.6a) to pinpoint our key designs: In first experiment, we *disabled grouped rollout*, but preserved an oversubscription strategy, i.e., feeding a lot prompts and harvest the first few ready responses. In this way the rollout easily bias to shorter responses. Consequently, the performance capped at less than 1 in validation score and stopped improving.

Then, we investigated the effect of fully on-policy mode. In ablation settings, we sort the batch *post hoc*, after all responses are generated. Here, we set the rollout batch size to 512 prompts, which is the number of prompts that will be consumed within 1 group in fully on-policy SortedRL. As a result, the oldest trajectory in this method is 4 times farther away from policy in training than other strategies. In contrast, in our approach, the prompts are consumed in 4 separate iteration, and trainer gets freshly generated responses in each step. The validation score shows that even with batch sorting, the off-policiness is holding back token-efficiency.

### 4.4.3 SENSITIVITY TO GROUPING SIZE

SortedRL introduced a new hyperparameter, group size $n$. This refers to the number of prompt batches to be loaded by the SortedRL controller every time the prompt pool clears. Let rollout batch size be $b$, then the total number of prompts to be loaded into rollout buffer is denoted as $nb$. The controller does not load new prompts until every sample in current buffer are fed to the trainer.

Therefore, a large $n$ causes the rollout engine to keep generate answers that are clustered in same length. An extreme case is infinitely big $n$, at which the trainer only get short data. As shown in

Fig.6b, it fails to improve. Similar degradation is also identified on $n = 8$. In contrast, $n = 2$ results in a data distribution near the baseline approach, and consequently lead to a baseline-like curve.

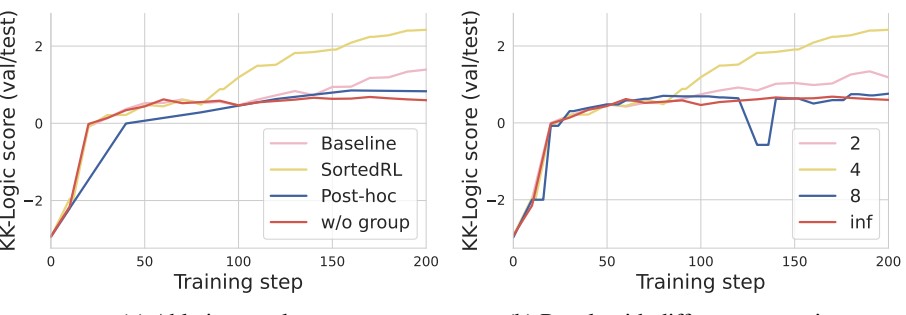

(a) Ablation results.      (b) Result with different group size.

Figure 6: Detailed Analysis.

## 5 RELATED WORK

### 5.1 RL TRAINING SYSTEMS

RLHF training frameworks have progressed from algorithm-centric libraries such as TRL (von Werra et al., 2020) and RL4LMs (Ramamurthy et al., 2022) to throughput-oriented systems like ColossalChat (Li et al., 2023), DeepSpeed-Chat with ZeRO (Rajbhandari et al., 2019; 2021), and NeMo-Aligner (Shen et al., 2024), which scale to thousands of GPUs. The newest entrants—OpenRLHF (Hu et al., 2024) and VeRL/HybridFlow (Sheng et al., 2024)—simplify RLHF for non-experts and offer "parallel-native" execution across heterogeneous hardware, supporting 3-D, ZeRO, and FSDP parallelism out of the box. However, none of these frameworks yet provides online batch scheduling or fine-grained rollout control, two capabilities that our work introduces.

### 5.2 LLM GENERATION OPTIMIZATION

Modern RLHF rollouts piggy-back on high-throughput LLM-serving stacks: both OpenRLHF and VeRL use vLLM's PagedAttention for rapid KV-cache access (Kwon et al., 2023), while VeRL can switch to SGLang's Radix Attention, which pins shared cache segments to avoid recomputation (Zheng et al., 2024). These engines pair optimized CUDA/HIP kernels with graph capture, speculative decoding, continuous batching from Orca (Yu et al., 2022), and chunked prefill from Sarathi (Agrawal et al., 2024). Yet they still target low-latency, online inference with frozen weights, whereas RLHF demands high-throughput, batched "semi-offline" generation whose weights shift after every policy update—changes that invalidate cached kernels, disrupt KV-cache layouts, and create rollout-to-update "bubbles," while magnifying sensitivity to speed–accuracy trade-offs such as quantization. *SortedRL* closes this gap by shrinking those bubbles and markedly boosting rollout throughput.

## 6 CONCLUSION

We propose *SortedRL*, an online length-aware scheduling scheme that delivers high hardware utilization and superior sample efficiency. Via fine-grained rollout control, SortedRL can reorder training batches online to construct sample-efficient micro-curriculum. In experiment verification, we observed robustly superior sample-efficiency with SortedRL on logical reasoning tasks and mathematical problems. With Qwen-2.5-32B and identical amount of training data, we achieved over 18% accuracy increment in AIME24 compared to baseline. Moreover, our sorted approach tackled the computation bubbles in rollout, cutting it from 74% down to less than 5.81%, bringing about up to near 40% boost in rollout throughput. SortedRL provides RL LLM training with two compute efficient methods with significantly less off-policy effect compared to canonical large batch training. At the same time, SortedRL also pronounced the importance of batch-wise normalization and selective batching, which might be a critical aspect in future research in LLM RL training algorithms.

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

# A DATASET EXAMPLE

```
Prompt
------
system
You are a helpful assistant. The assistant first thinks about the reasoning process
in the mind and then provides the user with the answer. The reasoning process and
answer are enclosed within <think> </think> and <answer> </answer> tags, respectively,
i.e., <think> reasoning process here </think><answer> answer here </answer>.
Now the user asks you to solve a logical reasoning problem. After thinking, when
you finally reach a conclusion, clearly state the identity of each character within
<answer> </answer> tags. i.e., <answer> (1) Zoey is a knight
(2)... </answer>.
<|im_end|>
<|im_start|>user
A very special island is inhabited only by knights and knaves. Knights always tell
the truth, and knaves always lie. You meet 3 inhabitants: Michael, Zoey, and Ethan.
Michael was heard saying, "Ethan is a knight if and only if Michael is a knight".
"Zoey is a knight or Ethan is a knight," Zoey mentioned. Ethan asserted:
"Michael is a knave if and only if Zoey is a knave".
So who is a knight and who is a knave?
<|im_end|>
<|im_start|>assistant
<think>assistant

Ground Truth
------
(1) Michael is a knight
(2) Zoey is a knight
(3) Ethan is a knight
```

Figure 7: Example (prompt, answer) pair for the LogicRL task.

```
Prompt
------
Solve the following math problem step by step. The last line of your
response should be of the form

    Answer: $Answer

(without quotes) where $Answer is the answer to the problem.

In triangle $ABC$, $\sin \angle A = \tfrac{4}{5}$ and $\angle A < 90^
\circ$.Let $D$ be a point outside triangle $ABC$ such that
$\angle BAD = \angle DAC$ and $\angle BDC = 90^\circ$.
Suppose that $AD = 1$ and that $\tfrac{BD}{CD} = \tfrac{3}{2}$.
If $AB + AC$ can be expressed in the form $\tfrac{a\sqrt{b}}{c}$ where
$a, b, c$ are pairwise relatively prime integers, find $a + b + c$.

Remember to put your answer on its own line after "Answer:".

Ground Truth
------
34
```

Figure 8: Example (prompt, answer) pair for the mathematical reasoning task.

# B ADDITIONAL ANALYSIS

Length is highly associated with the model performance. Reportedly in recent discoveries, reasoning capaility increment comes with increase response length. This aligns with the observations in our

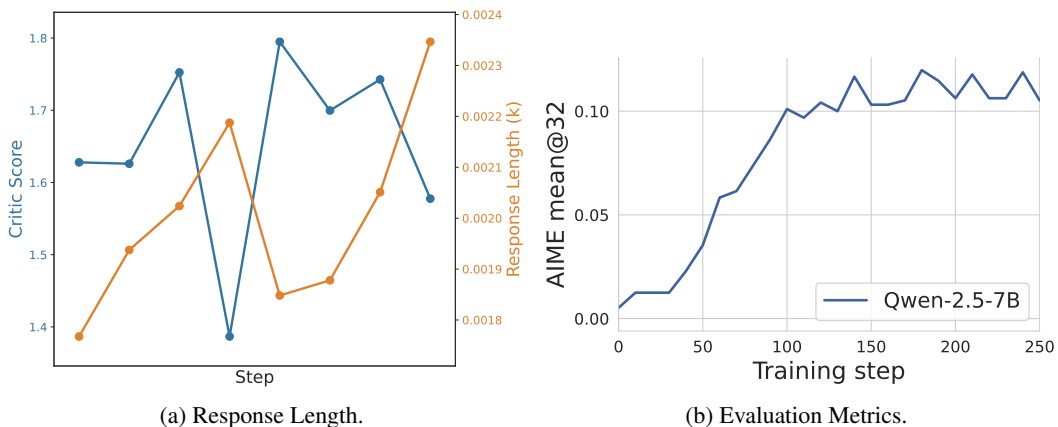

(a) Response Length.

(b) Evaluation Metrics.

Figure 9: Additional Analysis.

experiments (Fig.3b,4b). Given our length-sorted training schedule, we can have more insights into the role of response length in training. Fig.9a is a close-up look into two consecutive groups in SortedRL training. There is a clear short-short-short-long pattern in the rollout batches. This pattern forms a micro-curriculum for the training. The prompt remained in the buffer until clearing are generally questions require more reasoning steps to resolve.

Microscopically, longer responses tend to but not always have lower performance (Fig.9a). This can be partially explained by the fact that answers in longer sequences are more prone to be clipped. However, this low-score iteration is not harmful. Instead, the next iteration after long batch can achieve a higher score than previous short batches. Then the score keep drops as length increases until the next long batch concludes.

Macroscopically, we have some interesting observation in mathematical tasks. Just like local patterns mentioned earlier, the improvement of model performance are likely to show up on the falling edge of response length (310-330 steps, 340-350 steps of SortedRL; 480-550 in Baseline). Meanwhile, at comparable validation performance, SortedRL-tuned models has longer response length in both tasks.

