# OpenReview forum: "SortedRL: Accelerating RL Training for LLMs through Online Length-aware Scheduling"
_ICLR.cc/2026/Conference — Submitted to ICLR 2026_

### Official Review · Reviewer_3fnT · 2025-10-30

**Soundness:** 2
**Presentation:** 3
**Contribution:** 2
**Rating:** 4
**Confidence:** 3

**Summary:**

This paper presents SortedRL, an online length-aware scheduling strategy designed to address the efficiency bottleneck in reinforcement learning training for large language models. The core innovation is reordering rollout samples based on output lengths, prioritizing shorter samples for early updates while enabling large rollout batches and flexible update batches simultaneously. The method includes controllable off-policy training mechanisms and a co-designed RL infrastructure with a stateful controller and rollout buffer. Experiments on `LLaMA-3.1-8B` and `Qwen-2.5-32B` on logical reasoning and math reasoning tasks demonstrate significant reductions in bubble ratios (from $74\\%$ to $<6\\%$).

**Strengths:**

- **Important problem:** Addresses a real bottleneck in RL training for LLMs, where rollout can consume up to $70\\%$ of training time for long sequences (16k tokens)

- **Comprehensive system design:** Goes beyond algorithmic contribution to include infrastructure components (length-aware controller, stateful buffer) that enable the approach

- **Thorough ablation studies:** Figure 6 provides useful ablations showing the importance of grouped rollout and on-policy vs. post-hoc sorting

**Weaknesses:**

### Major Concerns

1. **Missing wall-clock time comparisons:** While Figures 3 & 4 show results vs. update steps, and Figure 5 shows throughput, the paper lacks end-to-end wall-clock time comparisons for the full training runs. Given that SortedRL introduces additional complexity (sorting, buffer management), it's crucial to see actual training time on the x-axis to understand real-world gains. The throughput gains in Figure 5 don't directly translate to total training time savings.

2. **Model selection concerns:** The choice of `Qwen-2.5-32B` for math experiments is problematic given documented data contamination issues for this model on math benchmarks (see https://arxiv.org/abs/2506.10947, https://arxiv.org/abs/2507.10532v1)
The paper only tests 2 base models (`LLaMA-3.1-8B`, `Qwen-2.5-32B`). Given that prompt/rollout length distributions can vary significantly across different models and tasks, more diverse model testing is needed to validate generalizability
For `Qwen-2.5-7B`, the authors note it shows limited test-time scaling (Fig 9b), but don't explore other 7B-scale alternatives.

3. **Tightly coupled design:** The ablation in Figure 6(a) shows that baseline actually outperforms individual components (Post-hoc, w/o group), suggesting all components must work together. This raises concerns about (1) Sensitivity to hyperparameters (2) Difficulty in adapting the method to different scenarios (3) Whether gains come from the scheduling strategy or from the specific combination of techniques

4. **Limited scope of experiments:** Only two task types (logical puzzles, math problems) are evaluated. Both tasks have rule-based evaluation - unclear if benefits transfer to tasks when generating a verdict is much more time-consuming such as LLM-as-a-judge.



### Technical Issues

5. **Grouped rollout details unclear:**
    1. "How many prompts will a batch include?" is not explicitly stated for all settings
    2. "How many batches might a single prompt's response be scattered into?" - this is critical for understanding off-policiness but not clearly explained
    3. The cache-aware loading policy ("no new prompts are loaded from the dataloader until all cached prompts have been consumed") needs more detail: what is the maximum queuing time or iterations?


6. **Prompt starvation mitigation:** Section 3.2 mentions preventing "starvation of certain prompts" by scavenging, but what if a prompt consistently generates very long responses (e.g., hard problems)? Its response would be scattered across many segments from different policy versions. The paper doesn't analyze the worst-case scenario or provide empirical data on how often this occurs and its impact on performance due to extremely off-policy.


7. **Batch normalization impact:** The paper claims selective batching is "particularly important for algorithms such as Reinforce++, where batch-wise normalization can substantially impact training outcomes" (line 242-243), but doesn't quantify or demonstrate this impact. What specifically makes the normalization sensitive to batch composition?


8. **Off-policy partial mode concerns:** Figure 3(b) shows a drastic, unstable explosion in response length for partial mode that leads to "unrecoverable degradation". The paper doesn't adequately explain why this happens or whether re-calculating importance sampling ratios might mitigate this issue.



### Minor Issues

- **Notation and clarity issues:** Equations (2) and (3): Many symbols lack explanation. Readers unfamiliar with PPO/Reinforce++ will struggle to understand these. The relationship between rollout batch size, update batch size, and group size could be explained more clearly upfront.

- **Possible typo in Section 4.3:** The text states "baseline, on-policy and partial SortedRL achieved 23.33%, 20.83%, and 19.69% accuracy" but then claims this follows "decreasing order of off-policiness (on-policy - partial SortedRL - baseline)". This seems backwards - if on-policy is least off-policy, it should have the best performance, but the numbers show baseline (23.33%) > on-policy (20.83%) > partial (19.69%). Please clarify.

**Questions:**

Please address my concerns in the above section. I don't have other questions.

---

### Official Review · Reviewer_EJ4u · 2025-10-31

**Soundness:** 1
**Presentation:** 1
**Contribution:** 1
**Rating:** 2
**Confidence:** 4

**Summary:**

This paper proposes a scheduling strategy in batch RL training systems to improve training efficiency. The core strategy is to start a large batch for generation and consume the batches according to the order of lengths. Experiments are taken on mathematical reasoning and logic domains. Experiment results show that the proposed strategy improves training speed of vanilla RL training.

**Strengths:**

1. The paper is well motivated as the bubble issue is a well-known issue in RLVR training.

**Weaknesses:**

1. Description of the central components of the proposed strategy is unclear, especially in Sec 3.1.
2. There is a lack of direct comparison with existing speedup techniques in RLVR training, including one-step-off RL training used in DeepCoder project [1] and fully asynchronous RL training in AReaL [2]. It is expected to have a comparison of the speedup with existing asynchronous training approaches.
3. The proposed strategy does not seem robust to the design choices, as evidenced by the collapse of partial mode in the logic RL experiment and failure of using a large batch size for rollout in the ablation study (Sec 4.4.3)

[1] Luo, M., Tan, S., Huang, R., Patel, A., Ariyak, A., Wu, Q., ... & Stoica, I. (2025). Deepcoder: A fully open-source 14b coder at o3-mini level. Notion Blog.
[2] Fu, W., Gao, J., Shen, X., Zhu, C., Mei, Z., He, C., ... & Wu, Y. (2025). AReaL: A Large-Scale Asynchronous Reinforcement Learning System for Language Reasoning. arXiv preprint arXiv:2505.24298.

**Questions:**

1. How does the proposed approach compare with asynchronous RL training approaches such as AReaL and one-step-off used in DeepCoder project in terms of training speed?
2. Why does the partial mode fails in logic RL experiment in Fig. 3? Considering the failure of partial mode, why not consider using the decoupled PPO objective in AReaL?
3. Could you provide experiment results showing the scaling property of the proposed strategy? That is, how would the training throughput change with different number of GPUs?

---

### Official Review · Reviewer_Qsod · 2025-11-02

**Soundness:** 3
**Presentation:** 3
**Contribution:** 3
**Rating:** 6
**Confidence:** 3

**Summary:**

This paper develops an online length-aware scheduling policy called SortedRL to improve rollout efficiency and maintain stability during training. It improves RL efficiency by sorting rollout samples by length for policy updates, thus directly tackling the rollout bottleneck in traditional RL algorithms.

Key components of SortedRL:
1. Online length aware scheduling: The controller sorts the incoming rollouts by lengths and updates the policy through early-termination once the update batch size has been reached. This leads to shorter sequences being prioritzed and over the course of a full rollout batch, a micro-curriculum is formed.
2. Controlled off-policyness: SortedRL supports two modes of operation: fully on-policy and partially on-policy. In the fully on-policy mode, fresh rollouts are generated for the cached prompts after each policy update. In the partial on-policy setting, caches the tokens and logits of incomplete rollouts and continues them after the policy update.

SortedRL results in an improved bubble ratio (74% in baseline to 6% in SortedRL) and attains superior performance over the baselines.

**Strengths:**

- SortedRL is a novel framework designed to alleviate the significant rollout bottleneck in RL and address the instability introduced by off-policy updates that come with large rollout batches.
- This system of sorting rollouts by output lengths for updates is intuitive and  improves both hardware efficiency (lower bubble ratio) and sample efficiency (improved performance at earlier steps) through a higher-degree of on-policyness.
 - The paper includes significant quantitative results to show the effectiveness of SortedRL on logic and math tasks and also provides an insight into the changes training dynamics. The ablations on grouped rollouts, fully on-policyness and groups size do a good job of furthering our understanding of this technique.
- The paper is for the most part clearly written and well-presented.

**Weaknesses:**

- A significant implicit assumption in the paper is that longer rollouts == harder prompts which is what enables the micro-curriculum. This largely holds true for math and reasoning tasks where longer rollouts mean longer, richer reasoning chains. However for other tasks like summarization, general instruction following, safety alignment etc. this is not necessarily true. The effectiveness of SortedRL on such tasks is unclear.
 - The paper would benefit from a deeper analysis into why SortedRL in the partial on-policy setting catastrophically failed so early on when training on LogicRL. This limits the usefulness of the partial on-policy mode and makes it difficult to use. Analyzing when the partial on-policy model should or should not be used would help with understanding and usability.
 - The new hyperparameter (group size) introduced in this work significantly impacts test time accuracy. The included ablation shows that larger group sizes result in imbalanced updates. However, it is not clear if or how the optimal group size changes with the training set. More analysis on group size and potentially some intuition or heuristic to determine an optimal group size would be welcome.

**Questions:**

The AIME24 numbers in lines 367-368 don't match those in Table 1. I am assuming the table is the source of truth but it would be good to correct it in the text.

---

### Official Review · Reviewer_shUf · 2025-11-12

**Soundness:** 1
**Presentation:** 1
**Contribution:** 2
**Rating:** 0
**Confidence:** 3

**Summary:**

This paper proposes a new RL post-training framework for language models, which introduces an online length-aware scheduling strategy that sorts rollout samples by length, prioritizing shorter generations. The method also includes a mechanism to manage partial trajectories, which can be fully on-policy (partial trajectories are discarded) or partial mode (partial trajectories are retained and resumed by the updated policy). The authors then conduct experiments on logical reasoning and math benchmarks and show the improved training efficiency, with relatively neutral performance gain.

**Strengths:**

- Significance: the paper addresses an important bottleneck in scaling RL training for reasoning models.
- Methodology: the proposed method, on a high level, is reasonable and easy-to-understand.

**Weaknesses:**

- Clarity: The method is described almost entirely through high-level qualitative descriptions. There are no formal algorithm blocks or pseudocode to define critical components like the "oversubscription strategy," "early termination" logic, or exactly how the "length-aware controller" manages the queue. This makes the mechanism ambiguous. Please provide some formal algorithm blocks in the paper to help people understand in details. Example: [[Phuong and Hutter, 2024](https://arxiv.org/pdf/2207.09238)].
- Reproducibility Issues: As noted, without precise algorithmic descriptions or provided code, it is unclear how practitioners can re-implement the proposed method. Please consider open-source the code or provide training details as clear as possible.
- Insufficient analysis of non-i.i.d. updates: The core idea relies on sorting data by length, which introduces a strong bias (non-i.i.d. batches) into the gradient updates (the "micro-curriculum"). The paper lacks a rigorous analysis or ablation study on how this specific bias affects the convergence, limiting the scalability of the proposed method.

**Questions:**

Please see the weakness section. I am happy to raise my score if the **reproducibility** concerns are substantially addressed. A method that cannot be replicated by the community cannot not provide the reliable knowledge necessary for conference acceptance.

---

### Meta-Review · Area_Chair_GW93 · 2026-01-06

**Summary:**

This submission receives ratings of 6, 4, 2, 0. The authors did not participate in the rebuttal phase. The major concerns raised by the reviewers are on the empirical demonstration of its proposed SortedRL framework, including its reproducibility and comparison against other async RL frameworks. All factors combined, this work is recommended with a rejection.

**Reviewer Concerns:**

No rebuttal is provided.

**Reviewer Scores:**

No rebuttal is provided.

---

### Decision · Program_Chairs · 2026-01-26

Reject